# VADB: A Large-Scale Video Aesthetic Database with Professional and Multi-Dimensional Annotations

**Qianqian Qiao**[*][1]**, Dandan Zheng**[2]**, Yihang Bo**[3]**, Bao Peng**[3]**, Heng Huang**[2]**,
Longteng Jiang**[2]**, Huaye Wang**[2]**, Jingdong Chen**[2]**, Jun Zhou**[2]**, and Xin Jin**[4]

[1]**Nanjing University**
[2]**Ant Group**
[3]**Beijing Film Academy**
[4]**Beijing Institute for General Artificial Intelligence**

## Abstract

Video aesthetic assessment, a vital area in multimedia computing, integrates computer vision with human cognition. Its progress is limited by the lack of standardized datasets and robust models, as the temporal dynamics of video and multimodal fusion challenges hinder direct application of image-based methods. This study introduces VADB, the largest video aesthetic database with 10,490 diverse videos annotated by 37 professionals across multiple aesthetic dimensions, including overall and attribute-specific aesthetic scores, rich language comments and objective tags. We propose VADB-Net, a dual-modal pretraining framework with a two-stage training strategy, which outperforms existing video quality assessment models in scoring tasks and supports downstream video aesthetic assessment tasks. The dataset and source code are available at `https://github.com/BestiVictory/VADB`.

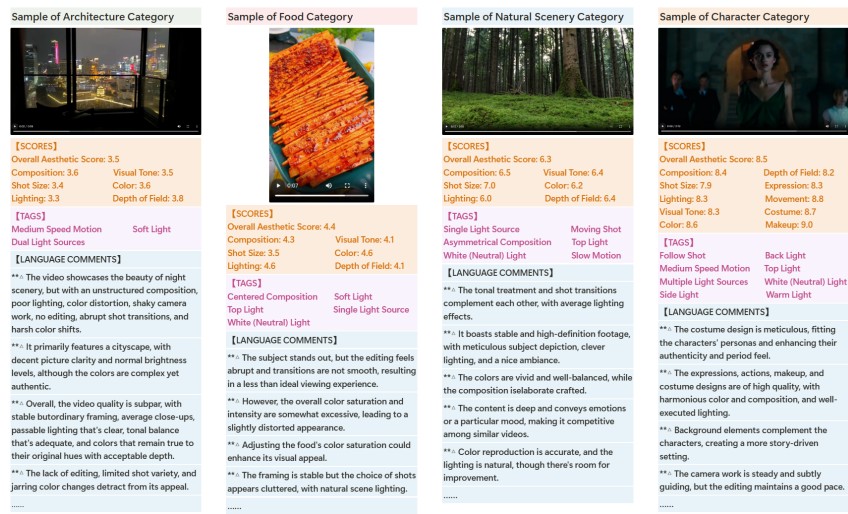

Figure 1: Dataset Examples.

*Corresponding author: Qianqian Qiao (602025320011@smail.nju.edu.cn)

39th Conference on Neural Information Processing Systems (NeurIPS 2025) Track on Datasets and Benchmarks.

# 1 Introduction

The rapid proliferation of short-video platforms and breakthroughs in generative AI have driven an unprecedented surge in internet video data, with millions of hours of content uploaded daily to major global platforms as reported by Statista[2]. Concurrently, user expectations for video content have evolved, extending beyond informational value to include aesthetic qualities such as visual composition, color harmony, and dynamic rhythm. This interplay of supply and demand has elevated video aesthetic assessment as a pivotal research area in both academia and industry, highlighting its increasing importance and urgency.

To address this, the development of a scientific and systematic video aesthetic evaluation framework, the creation of a high-quality, multidimensional video aesthetic dataset with rich annotations, and the design of robust, generalizable, and accurate assessment models are critical research objectives. Unlike image aesthetic assessment, which leverages established principles like the golden ratio and color harmony [Yi Lu, 2021][Houlgate, 2009], video aesthetic assessment is more complex due to its dynamic spatiotemporal nature. A comprehensive evaluation system thus requires thorough investigation of key aesthetic attributes shaping viewer perception and the establishment of tailored standards for diverse video genres. Methodologically, integrating objective technical metrics—such as camera movement and composition types—with traditional subjective evaluations is essential for a balanced assessment approach.

An ideal video aesthetic dataset should be grounded in a systematic evaluation framework with consistent annotation protocols, supported by professional annotators with deep aesthetic expertise and video production experience to ensure both quality and scalability. Annotation richness is equally vital, encompassing fine-grained scores, tag-based labels, and language comments to clarify scoring rationales and enable models to capture nuanced aesthetic reasoning.

While image aesthetic assessment has advanced significantly in areas like classification, score distribution, and attribute analysis [Jin et al., 2023][Jin et al., 2024a][Huang et al., 2024], video aesthetic assessment lags due to technical challenges and the scarcity of large-scale datasets. Notably, the pretrained CLIP model[Radford et al., 2021] has shown exceptional performance in image aesthetic assessment[Sheng et al., 2023][Jin et al., 2024b] and video retrieval[Luo et al., 2021]. Utilizing multimodal video aesthetic datasets with detailed language annotations for CLIP-based pretraining offers substantial potential to overcome current limitations in video aesthetic research.

This study addresses these challenges through the following contributions:

**A set of video aesthetic annotation criteria.** A detailed framework developed by a team of film and television professionals, outlining the scoring criteria for an overall aesthetic score, 10 specific attribute scores[3], and selection guidelines for 34 technical tags[4].

**Video Aesthetic Database (VADB).** The largest dataset of its kind, comprising 10,490 videos, each labeled by at least 13 professional annotators, all with 7 or 11 scores, an average of 22 language comments and 7 objective tags. (**Note:** 80% of VADB is available at `https://github.com/BestiVictory/VADB`, while the remaining 20% contains protected materials.)

**A novel video aesthetic assessment model (VADB-Net).** VADB-Net based on a multimodal CLIP framework, delivering superior scoring performance.

# 2 Related Work

## 2.1 Video Aesthetic Dataset

To highlight the limitations of video aesthetic datasets, we first review the advancements in video quality datasets as a benchmark. Video quality datasets have progressed significantly, evolving from laboratory-controlled compression artifacts [Wang et al., 2016, 2017] to real-world scenarios with mixed distortions [Hosu et al., 2017, Sinno and Bovik, 2019]. Annotation methodologies have advanced from single-score ratings [Sinno and Bovik, 2019] to multi-dimensional quality attributes and textual descriptions [Hosu et al., 2017, Duan et al., 2024]. Concurrently, dataset scales have increased from hundreds of clips [Li et al., 2020] to tens of thousands, with annotations reaching millions [Ying et al., 2021, Duan et al., 2024].

---

[2]https://www.statista.com/

[3]Scoring Criteria and Example Videos of VADB

[4]Tag Annotation Criteria and Example Videos of VADB

Table 1: 10 Aesthetic Attributes of VADB

| Type | Attribute | Interpretation |
|------|-----------|----------------|
| General | Composition (Com) | Evaluates whether the layout of visual elements is harmonious, with the subject prominently featured, avoiding clutter or unbalanced focus. |
| | Shot Size (SS) | Identifies the framing range (e.g., wide, medium, or close shot) and assesses its suitability for conveying the intended content. |
| | Lighting (Lig) | Analyzes whether light clearly highlights the subject, avoiding issues like underexposure, overexposure, or distracting shadows. |
| | Visual Tone (V&T) | Examines the overall brightness, darkness, or color temperature of the image and its alignment with the content's mood. |
| | Color (Col) | Assesses whether colors appear natural or stylized, ensuring no distortion or oversaturation detracts from the viewing experience. |
| | Depth of Field (D&F) | Determines if the depth of field, including background blur, is appropriate to highlight the subject without obscurity or background dominance. |
| Character -specific | Expression (Exp) | Captures whether the character's demeanor is natural and vivid or conveys the intended emotion. |
| | Movement (Mov) | Evaluates whether actions are clear and fluid, and contextually appropriate. |
| | Costume(Cos) | Checks if attire is contextually appropriate for the scene. |
| | Makeup (Mak) | Assesses whether makeup is suitable, avoiding unnatural or discordant appearances. |

By contrast, video aesthetic quality datasets have developed more slowly, constrained by challenges in annotation standardization and high labeling costs. Early efforts, such as the Telefonica dataset [Bylinskii et al., 2015] with 160 YouTube videos annotated using five-level aesthetic ratings, established initial benchmarks. Subsequent datasets introduced greater diversity, including Niu et al.'s 2,000 professional and amateur videos with binary classification [Niu and Liu, 2012] and the VAQ700 dataset with 700 daily-life videos and multi-annotator ratings [Tzelepis et al., 2016]. Larger-scale datasets, such as AVAQ6000 [Kuang et al., 2019] with 6,000 drone videos, relied on binary labels, limiting their annotation richness. More recently, DIVIDE-3k [Wu et al., 2023] provided multi-dimensional ratings for 3,590 videos. Despite these advances, video aesthetic datasets remain constrained by limited scale, inconsistent annotations, and insufficient dimensional richness compared to video quality datasets, indicating a need for more comprehensive solutions.

## 2.2 Video Aesthetic Assessment Methods

Traditional methods rely on photographic principles, using handcrafted features to evaluate visual appeal. [Luo and Tang, 2008] extracted subject regions and motion features to distinguish professional from amateur videos. [Moorthy et al., 2010] utilized features like motion amplitude ratio and adherence to the rule of thirds, pooling frame-level features into video-level representations. [Niu and Liu, 2012] combined static image aesthetic features with dynamic features, for multi-class quality classification in professional video production. [Yeh et al., 2013] introduced motion features derived from optical flow within a temporal-aware framework. [Peng et al., 2021] evaluated robotic dance aesthetics by integrating spatial and shape features. These methods, while effective, are limited in capturing complex dynamic patterns.

Deep learning approaches, though constrained by limited video aesthetic datasets, show significant promise. [Phatak et al., 2019] applied CNNs to assess the impact of motion speed on aesthetics, outperforming handcrafted features. [Kuang et al., 2020] proposed a multimodal CNN that fused drone video, trajectory, and 3D structural features for aesthetic classification. [Asarkar and Phatak, 2019] integrated color contrast features with deep learning to explore connections between video emotion and aesthetics. [Wu et al., 2023] introduced the DOVER video quality assessor, combining aesthetic and technical perspectives to evaluate user-generated content (UGC) video quality. Despite fewer studies, these advancements highlight deep learning's potential to enhance aesthetic assessment.

## 2.3 Learning Visual Representations from Text Supervision

Text-supervised visual representation learning, leveraging abundant internet image-text pairs, has become a prominent research area. CLIP [Radford et al., 2021] learns robust visual representations through contrastive language-image pre-training, enabling effective transfer to tasks like image retrieval [Baldrati et al., 2023] and aesthetic assessment [Sheng et al., 2023, Jin et al., 2024b].

In video research, ClipBERT [Lei et al., 2021] adapts image-text pre-training for video question answering and text-to-video retrieval, while Clip4Clip [Luo et al., 2021] and X-CLIP [Ni et al., 2022] extend this approach to video-text pre-training, improving video retrieval performance. Inspired by these approaches, VADB-Net applies CLIP's representations to video aesthetic assessment.

## 3 VADB

### 3.1 Video Categories & Attributes & Tags

VADB categorizes videos into four types-character, natural scenery, architecture, and food-with character videos predominant, comprising 8,130 segments. Videos, sourced from documentaries, films, TV dramas, variety shows, news, user-generated content, and AIGC material, have durations of 5–20 seconds to balance content diversity and processing efficiency. This design ensures diverse sources and visual quality, ranging from professional to amateur, enhancing dataset generalizability.

The attributes listed in Table 1 are grounded in the aesthetic framework of film and television studies [Wu, 2024, Matbouly, 2022, Petrogianni et al., 2022, Arijon, 2013], enabling a comprehensive evaluation of a video's aesthetic quality. Notably, given the dynamic nature of subjects and the richness of emotional expression in character videos, specific character-specific attributes are annotated in addition to general attributes.

The tag annotation categories are shown in Figure 4. Tags are derived from three technical factors—camera movement, composition, and lighting—that influence video aesthetics. Camera movement affects dynamic visual expression, composition determines visual balance and hierarchy, and lighting shapes mood and texture. By focusing on observable visual elements, this approach reduces subjective biases, while concise and precise tags simplify annotation complexity.

### 3.2 Annotation Criteria

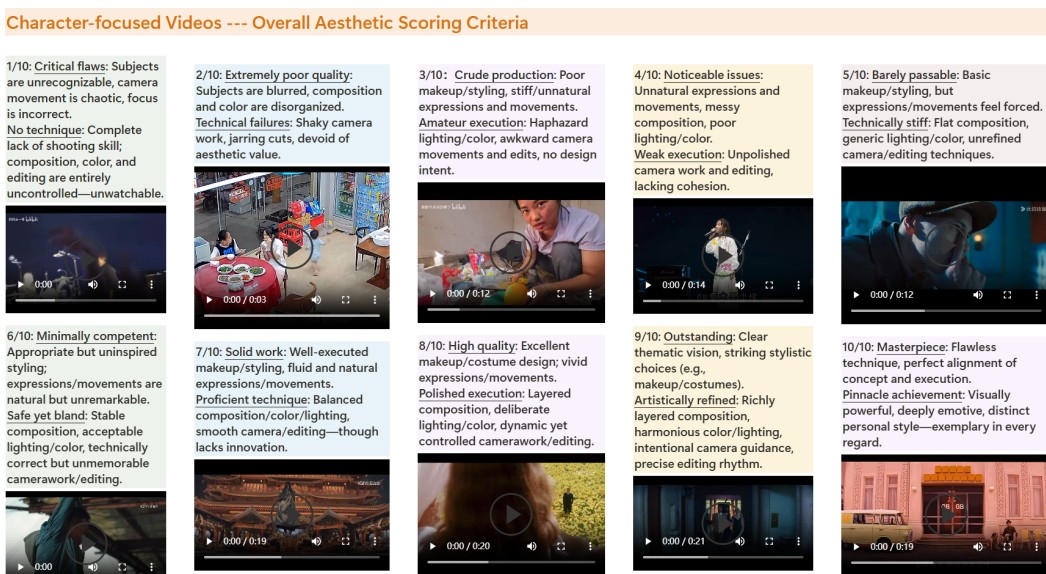

Figure 2: Annotation guidelines and example videos for aesthetic scoring of character videos: 1-3 show significant technical and aesthetic flaws; 4-5 meet basic standards with evident weaknesses; 6-7 meet standards with average execution; 8-9 exhibit technical skill and artistic merit; 10 reflect exceptional technical and artistic integration.

Annotation criteria, including scoring criteria and tag guidelines, were developed by experts from the Beijing Film Academy, drawing on their their rich experience in film and television production and teaching accumulation. The expert team thoroughly considered the diverse video aesthetic quality within the VADB, ranging from low-quality casual daily footage to high-quality cinematic works, to ensure comprehensive coverage of the annotation criteria.

To minimize the influence of individual aesthetic preferences, we established standardized guidelines for overall and attribute-specific video scoring across categories, with score ranges defined by textual descriptions and example videos. The scoring criteria adopt a progressive assessment model to form a

logical and hierarchical video aesthetics evaluation system. Additionally, to facilitate tag annotation, comprehensive textual explanations and illustrative videos were provided for various tag types.

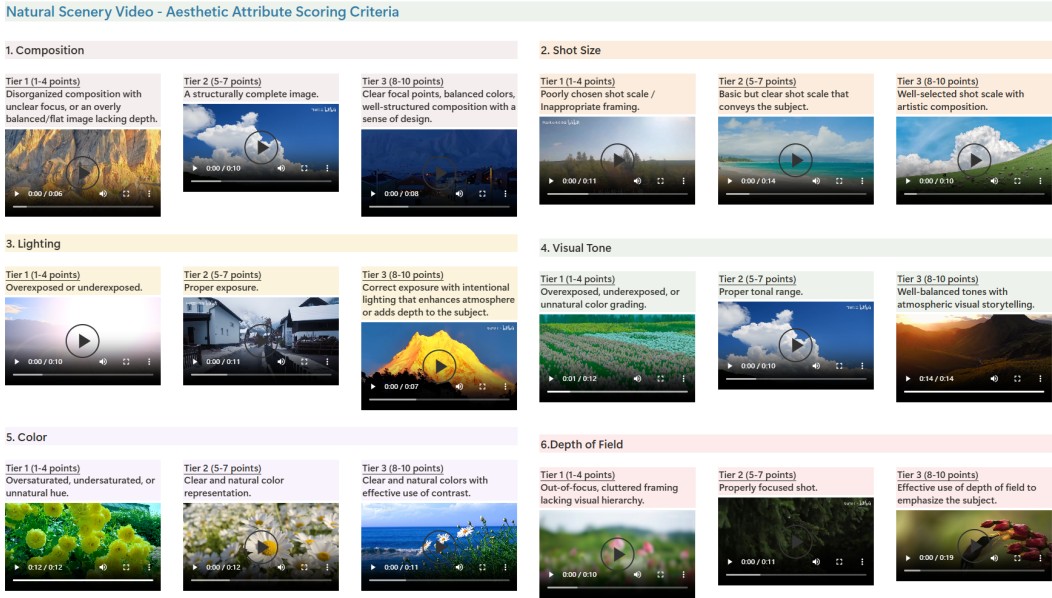

Figure 3: Scoring criteria and example videos for the aesthetic attributes of natural scenery videos. Additionally, we provided similarly detailed scoring criteria for the aesthetic attributes of character, food, and architecture videos. Example videos were selected by the expert team to illustrate the characteristics of each score range, with at least three videos of different scenarios for each range, enabling annotators to intuitively understand the evaluation criteria.

| Tag Title | Tag Category | Example of explanation |
|---|---|---|
| **Basic Camera Movements** | Push-in Shot | The camera moves toward the subject, or the lens focal length is adjusted to make the frame gradually approach the subject from a distance. |
| | Fixed Shot; Pull-back Shot; Pan Shot; Moving Shot; Follow Shot; Jib Up Shot; Jib Down Shot | |
| **Camera Movement Speed** | Slow Motion | The subject or background changes slowly in the frame, giving the audience enough time to observe details. |
| | Medium Speed Motion; Fast Motion | |
| **Composition Types** | Rule of Thirds Composition | The frame is divided into three equal parts, with key visual elements placed at the intersections or along the lines to create a more layered composition. |
| | Symmetrical Composition; Asymmetrical Composition; Centered Composition; Framing Composition; Leading Lines Composition | |
| **Number of Light Sources** | Single Light Source; Dual Light Sources; Multiple Light Sources | |
| **Light Source Position** | Back Light | The light shines from behind the subject, creating a glowing outline; enhancing depth and dimensionality. Ideal for separating the subject from the background and creating silhouette effects. |
| | Front-Side Light; Side Light; Bottom Light; Top Light; Front Light; Back-Side Light | |
| **Light Quality** | Hard Light | Strong, direct light with sharp shadows and high contrast. |
| | Soft Light; Diffused Light | |
| **Light Color** | White (Neutral) Light | Natural, neutral lighting that accurately represents the true colors of objects. |
| | Warm Light; Cool Light; Colored Light | |

Figure 4: Annotators label only the Tag Category, guided by a standardized document with explanations and example videos.

### 3.3 Annotation Team

We collaborated with the Beijing Film Academy to form a professional annotation team of 37 members, led by senior professors in film and television studies who oversaw personnel recruitment, training, and quality control of the annotation process. During recruitment, the following eligibility criteria were established:

1) At least three years of experience in film and television art appreciation;

2) A professional background in film and media studies;

3) A bachelor's degree or higher;

4) Availability to commit to a full month of continuous annotation work.

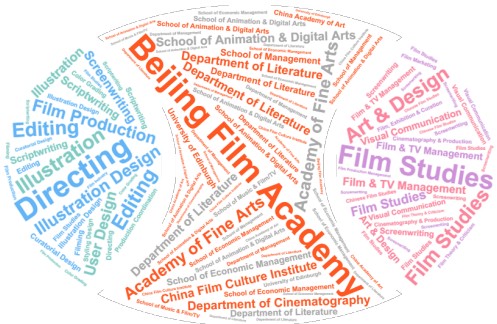
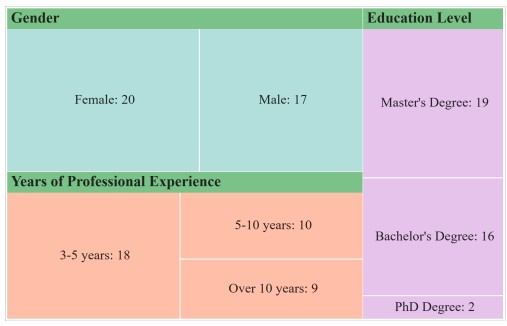

(a) Three word clouds (left to right) depict the annotation team's expertise, affiliations, and study fields. 68% of the team have film/TV tech experience, 43% in movie production, 14% in film theory research. Team covers film creation, research, and industry practice, with diverse complementary skills.

(b) The team maintains a balanced gender ratio, exceptional academic qualifications, and extensive professional experience. The combination of advanced education and seasoned expertise ensures the professionalism and reliability of the annotation process.

Figure 5: Composition and qualifications of the annotation team

It should be noted that prior to the commencement of the annotation activity, all annotators were fully informed of the potential risks involved (including psychological and emotional stress, time conflicts, and possible platform malfunctions), ensuring that their participation was entirely voluntary. In addition, multiple measures were implemented to safeguard their legal rights and interests. Furthermore, this annotation activity has been formally approved by the Institutional Review Board (IRB) of Beijing Electronic Science and Technology Institute, and the approval process complies with established academic ethical standards.

## 3.4 Annotation Process

Before initiating annotation, the team leader conducted systematic training on annotation criteria, combining theoretical explanations with case demonstrations to ensure team members thoroughly understood the core principles and detailed requirements. The training aimed to enable accurate data annotation according to unified standards.

To maintain quality during annotation, a quality inspection team, led by the team leader and comprising three highly experienced members, was established to review annotation quality. Substandard annotations were promptly returned for revision. The inspection team analyzed common issues, provided feedback through representative case studies, and addressed annotators' queries in real time. This continuous collaboration enhanced the accuracy and efficiency of the annotation process, ensuring the project's high-quality completion.

The study employed a self-built annotation platform for annotation tasks. A total of 13,000 videos were uploaded to the backend, with a video aesthetic annotation system designed to include three tasks: scoring (1-10 single-choice scale), commenting (free-text format), and tagging (multiple-choice mode). Leveraging the concept of "collective intelligence" for accurate annotation, each video required 10 annotators for scoring and commenting and 3 for tagging, resulting in at least 13 annotations per video. Task assignments were personalized based on annotators' preferences and availability, with dynamic adjustments made according to real-time progress. The entire annotation process was completed in approximately 35 days.

Video annotation compensation is divided into two standards based on task complexity and workload: (1) annotations involving scoring and commenting, with an average payment of 1.5 CNY per task; and (2) annotations requiring only tag labeling, with an average payment of 0.5 CNY per task. These standards ensure professionalism and fair incentivization.

## 3.5 Cleaning of Labeled Data

As shown in Table 2, we employed Krippendorff's Alpha (ordinal) coefficient to quantitatively assess the annotation consistency across dimensions. Results show that most coefficients range from 0.55 to 0.66, with relatively high consistency observed in dimensions like makeup and expression. Meanwhile, we note that consistency in some dimensions still has room for improvement, which may be attributed to the inherent subjectivity of aesthetic evaluation.

Table 2: Krippendorff's Alpha Coefficients for Each Annotation Dimension

| Dimension | Overall | V&T | SS | D&F | Lig | Com | Col | Mov | Mak | Cos | Exp |
|---|---|---|---|---|---|---|---|---|---|---|---|
| Alpha | 0.59 | 0.57 | 0.61 | 0.55 | 0.57 | 0.61 | 0.54 | 0.63 | 0.58 | 0.59 | 0.66 |

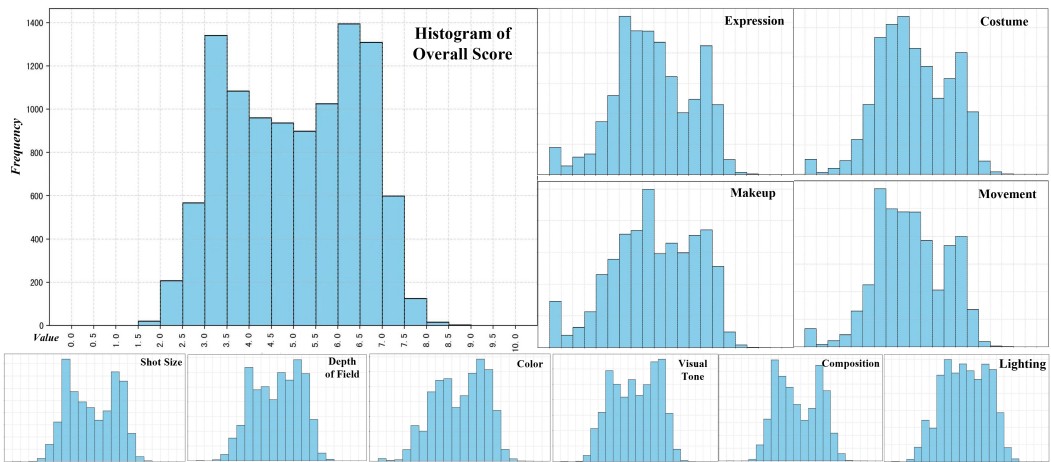

Figure 6: Histogram of overall and attribute score distributions, showing a dense mid-range and sparse extremes, consistent with typical rating patterns.

After cleaning the 13,000 annotated videos, 10,490 valid entries were retained. The cleaning process involved the following steps:

1) For aesthetic scores, exclude videos labeled by fewer than 5 annotators. Remove ratings with squared difference from the mean exceeding 8 as outliers. If a video's max-min score difference exceeded 5, discard ratings significantly deviating from the mean.

2) For tags, videos with only two tags and low-frequency tags appearing once were removed.

3) For comments, highly similar texts were elimina ted, retaining only those linked to valid videos.

This process ensured high-quality standards for scores, tags, and comments, establishing a robust foundation for subsequent research.

## 4 VADB-Net

### 4.1 Pre-training Stage

The Video Encoder, built upon CLIP ViT-B/32[Radford et al., 2021], employs 3D convolution initialization to extend 2D convolutional kernels for extracting spatiotemporal features. After 12-layer Transformer encoding, frame-level mean pooling generates fixed-length video representations.

The model adopts a dual-text encoder structure for distinct text inputs. The primary encoder retains CLIP's 12-layer Transformer architecture, processing natural language comments. An independent tag encoder (sharing the visual encoder's architecture but with separate parameters) encodes aesthetic tags like "symmetric composition" or "top lighting." Both produce 512-dimensional features, forming a complementary text representation system.

Dynamic Fusion Module integrates the two text features via a learnable attention mechanism. This module computes a dynamic weight $\alpha$, with the fused feature defined as $f_{\text{fused}} = \alpha \cdot f_{\text{Comm}} + (1 - \alpha) \cdot f_{\text{Tag}}$, where $\alpha$ is normalized via softmax. Initialized at 0.7, $\alpha$ balances the dominance of natural language descriptions while adaptively adjusting contribution of aesthetic tags based on input content.

Similarity computation employs a symmetric contrastive loss strategy. A learnable temperature parameter adjusts the cosine similarity scale, and matrix multiplication is performed between fused text and video features. Bidirectional cross-entropy loss ensures feature space alignment, preserving CLIP's normalized projection space properties while enhancing enhancing aesthetic feature representation.

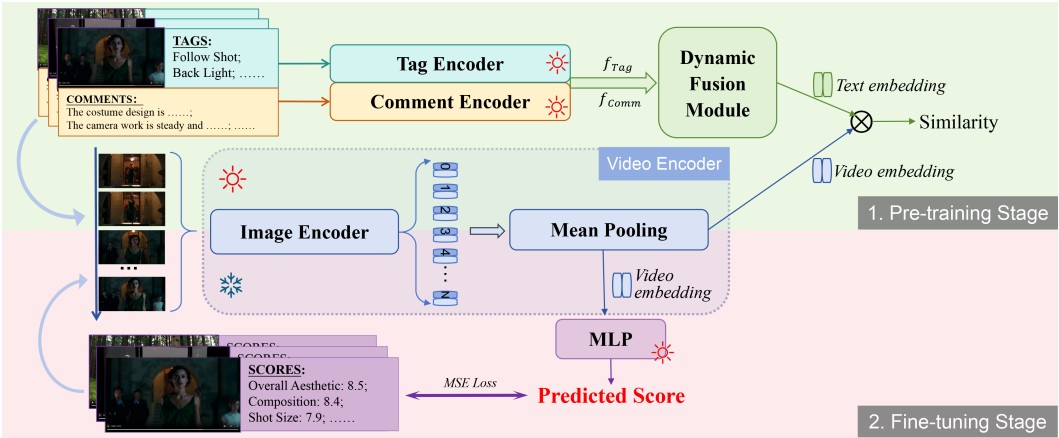

Figure 7: VADB-Net employs a two-stage training strategy. In the Pre-training Stage, the video encoder extracts frame sequence features, while dual text encoders process language comments and aesthetic tags. Dynamic Fusion Module adaptively integrates text features, and a symmetric contrastive loss aligns the video-text feature spaces to pre-train the Video Encoder. In the Fine-tuning Stage, the pre-trained Video Encoder extracts video embeddings, and an MLP regression network is trained to predict aesthetic scores.

Training was conducted on four NVIDIA H20 GPUs in parallel, using 226,940 video-comment-tag samples. Inputs consisted of 12 uniformly sampled video frames (1 fps) and 32-token text sequences. Optimization settings included an initial learning rate of 0.0001, a 10% warmup ratio, a 0.9 decay rate, and a batch size of 64, with training spanning two epochs.

## 4.2 Fine-tuning Stage

In this stage, the Video Encoder's parameters are frozen to preserve its pre-trained visual representation capabilities. Input videos are processed by the encoder to generate frame-level feature sequences, which are aggregated into a 512-dimensional video-level global representation through mean pooling along the temporal dimension.

A lightweight MLP regression network is then constructed: the first maps 512-dimensional features to 512 dimensions with ReLU activation, the second reduces to 256 dimensions with ReLU, and the third linearly projects to a 1-dimensional aesthetic score. The model is trained using MSE loss to minimize the discrepancy between predicted and ground-truth scores. By freezing the backbone network and fine-tuning only the top MLP layers, the approach ensures efficient task adaptation while retaining the pre-trained model's feature extraction capabilities.

For training setups, a single NVIDIA H20 GPU is utilized, with the VADB split into training and validation sets at a 4:1 ratio. Model optimization is performed with a learning rate of 0.001. Detailed training details will be presented in the supplementary material.

## 4.3 Experiment

### 4.3.1 Ablation Experiment

Based on the original two-stage training framework of VADB-Net, we designed two sets of ablation experiments to verify the effectiveness of core components:

**Ablation Experiment 1 (Without Pretrained Encoder):** The pretraining process in the first stage is removed, and the untrained CLIP ViT-B/32 encoder is directly used as the video feature extractor. Only in the second stage, the MLP regression network is trained on the VADB dataset to predict aesthetic scores. This setup is used to verify the impact of the pretraining stage on the model's feature extraction capability.

**Ablation Experiment 2 (Simplified Fine-tuning Layer):** The encoder pretrained on VADB in the first stage is retained, but the MLP regression network in the second stage is replaced with a simple linear layer (without hidden layers and activation functions), and the score is predicted only through linear mapping. This setup is used to verify the impact of the complexity of the network structure in the fine-tuning stage on the scoring accuracy.

All ablation experiments are conducted on the same training/validation set of the VADB dataset. The experimental results shown in Table 3 effectively verify the effectiveness of both stages.

Table 3: Results of ablation experiments

| Metric | Model | Dimension | | | | | | | | | | |
|---|---|---|---|---|---|---|---|---|---|---|---|---|
| | | Overall | V&T | SS | D&F | Lig | Com | Col | Mov | Mak | Cos | Exp |
| SRCC↑ | Ablation 1 | 0.84 | 0.87 | 0.85 | 0.84 | 0.83 | 0.77 | 0.79 | 0.64 | 0.60 | 0.65 | 0.61 |
| | Ablation 2 | 0.90 | 0.84 | 0.85 | 0.84 | 0.82 | 0.79 | 0.83 | 0.83 | 0.82 | 0.78 | 0.78 |
| | VADB-Net | 0.93 | 0.92 | 0.93 | 0.90 | 0.92 | 0.93 | 0.89 | 0.91 | 0.87 | 0.91 | 0.90 |
| PLCC↑ | Ablation 1 | 0.85 | 0.88 | 0.87 | 0.83 | 0.83 | 0.77 | 0.78 | 0.67 | 0.62 | 0.66 | 0.61 |
| | Ablation 2 | 0.90 | 0.85 | 0.85 | 0.84 | 0.81 | 0.78 | 0.82 | 0.80 | 0.79 | 0.76 | 0.74 |
| | VADB-Net | 0.93 | 0.92 | 0.93 | 0.90 | 0.92 | 0.94 | 0.88 | 0.89 | 0.84 | 0.91 | 0.89 |
| RMSE↓ | Ablation 1 | 0.59 | 0.45 | 0.51 | 0.68 | 0.73 | 1.01 | 0.83 | 1.6 | 1.4 | 1.3 | 2.1 |
| | Ablation 2 | 0.51 | 0.75 | 0.80 | 0.84 | 0.90 | 1.09 | 0.80 | 1.39 | 1.20 | 1.26 | 1.83 |
| | VADB-Net | 0.54 | 0.62 | 0.51 | 0.66 | 0.59 | 0.52 | 0.74 | 0.69 | 1.00 | 0.66 | 0.78 |
| KRCC↑ | Ablation 1 | 0.65 | 0.67 | 0.65 | 0.64 | 0.63 | 0.58 | 0.59 | 0.46 | 0.42 | 0.47 | 0.44 |
| | Ablation 2 | 0.72 | 0.64 | 0.65 | 0.65 | 0.65 | 0.60 | 0.64 | 0.63 | 0.62 | 0.59 | 0.58 |
| | VADB-Net | 0.77 | 0.75 | 0.76 | 0.73 | 0.76 | 0.77 | 0.72 | 0.75 | 0.70 | 0.75 | 0.73 |

### 4.3.2 Comparative Experiment

Although video aesthetic scoring models are scarce in academia, recent progress in video quality assessment has been notable. The FAST-VQA framework [Wu et al., 2022] efficiently learns quality-related features through end-to-end training, while SimpleVQA [Sun et al., 2022] effectively extracts spatial and temporal features. Furthermore, ModularBVQA [Wen et al., 2024] achieves accurate predictions by jointly processing visual content, resolution, and frame rate. Table 4 presents the comparative experimental results between VADB-Net and these three models.

All models were evaluated on the VADB dataset using consistent training and validation sets. Experimental results show that the proposed VADB-Net surpasses existing video quality assessment models in video aesthetic quality assessment. Moreover, the Video Encoder, pretrained in the initial phase of this study, demonstrates robust performance in aesthetic scoring and versatility for downstream tasks, such as aesthetic classification and score distribution prediction.

Table 4: Comparison of SimpleVQA, FastVQA, ModularBVQA and VADB-Net on VADB.

| Metric | Model | Dimension | | | | | | | | | | |
|---|---|---|---|---|---|---|---|---|---|---|---|---|
| | | Overall | V&T | SS | D&F | Lig | Com | Col | Mov | Mak | Cos | Exp |
| SRCC↑ | SimpleVQA | 0.85 | 0.82 | 0.85 | 0.79 | 0.83 | 0.86 | 0.77 | 0.81 | 0.79 | 0.82 | 0.81 |
| | FastVQA | 0.92 | 0.91 | 0.91 | 0.90 | 0.92 | 0.93 | 0.89 | 0.77 | 0.84 | 0.87 | 0.87 |
| | ModularBVQA | 0.89 | 0.88 | 0.90 | 0.86 | 0.89 | 0.90 | 0.84 | 0.87 | 0.82 | 0.87 | 0.82 |
| | VADB-Net | 0.93 | 0.92 | 0.93 | 0.90 | 0.92 | 0.93 | 0.89 | 0.91 | 0.87 | 0.91 | 0.90 |
| PLCC↑ | SimpleVQA | 0.86 | 0.82 | 0.87 | 0.79 | 0.83 | 0.88 | 0.77 | 0.81 | 0.77 | 0.83 | 0.82 |
| | FastVQA | 0.93 | 0.91 | 0.93 | 0.90 | 0.92 | 0.94 | 0.89 | 0.77 | 0.83 | 0.88 | 0.88 |
| | ModularBVQA | 0.89 | 0.88 | 0.92 | 0.85 | 0.89 | 0.92 | 0.84 | 0.87 | 0.81 | 0.89 | 0.81 |
| | VADB-Net | 0.93 | 0.92 | 0.93 | 0.90 | 0.92 | 0.94 | 0.88 | 0.89 | 0.84 | 0.91 | 0.89 |
| RMSE↓ | SimpleVQA | 0.75 | 0.88 | 0.72 | 0.90 | 0.85 | 0.68 | 1.01 | 0.88 | 1.16 | 0.85 | 0.97 |
| | FastVQA | 0.65 | 0.72 | 0.64 | 0.73 | 0.66 | 0.56 | 0.81 | 1.18 | 1.15 | 0.87 | 0.91 |
| | ModularBVQA | 0.65 | 0.74 | 0.60 | 0.78 | 0.70 | 0.57 | 0.85 | 0.75 | 1.08 | 0.70 | 1.08 |
| | VADB-Net | 0.54 | 0.62 | 0.51 | 0.66 | 0.59 | 0.52 | 0.74 | 0.69 | 1.00 | 0.66 | 0.78 |
| KRCC↑ | SimpleVQA | 0.65 | 0.62 | 0.65 | 0.59 | 0.63 | 0.66 | 0.57 | 0.62 | 0.60 | 0.63 | 0.62 |
| | FastVQA | 0.76 | 0.74 | 0.74 | 0.73 | 0.77 | 0.77 | 0.72 | 0.56 | 0.66 | 0.71 | 0.69 |
| | ModularBVQA | 0.71 | 0.70 | 0.71 | 0.67 | 0.71 | 0.71 | 0.66 | 0.69 | 0.65 | 0.70 | 0.65 |
| | VADB-Net | 0.77 | 0.75 | 0.76 | 0.73 | 0.76 | 0.77 | 0.72 | 0.75 | 0.70 | 0.75 | 0.73 |

### 4.3.3 Statistical Significance

Due to space constraints in the paper, we only present the performance metrics and statistical validation results of the overall score prediction branch in VADB-Net, as shown in Table 5.

Table 5: Performance metrics and statistical validation of overall score prediction branch

| Index | Value | P-value | 95% Confidence Interval | Error Bars (Lower/Upper) |
|---|---|---|---|---|
| MSE | 0.2941 | – | [0.2718, 0.3186] | (0.0224, 0.0244) |
| SROCC | 0.9299 | $< 0.001$ | [0.9232, 0.9353] | (0.0067, 0.0054) |
| PLCC | 0.9305 | $< 0.001$ | [0.9242, 0.9372] | (0.0063, 0.0067) |
| KRCC | 0.7704 | $< 0.001$ | [0.7602, 0.7811] | (0.0101, 0.0107) |
| Binary ACC | 0.9180 | $< 0.001$ | [0.9061, 0.9295] | (0.0119, 0.0114) |

## 5 Broader Impacts

While the proposed VADB dataset and VADB-Net model advance research in video aesthetic assessment, their potential social and ethical implications should also be acknowledged.

(1) The aesthetic annotation standards developed in this study are not intended as universal guidelines, but are applicable only to datasets with similar characteristics. Overgeneralization may lead to cross-cultural misinterpretations and cognitive biases, potentially weakening the representation of non-mainstream cultural expressions.

(2) Although the annotation team has strong professional expertise, all members are from the Beijing Film Academy, and their aesthetic judgments are shaped by specific cultural and educational backgrounds. This may introduce a single-cultural bias into the dataset. Models trained on such data could inadvertently reinforce these biases as universal norms, limiting cultural diversity and undervaluing alternative creative expressions.

(3) The dataset's focus on human-centered videos may further induce a human-centric aesthetic bias, reducing the model's ability to evaluate other video types. Deployed on content platforms, this could unintentionally diminish non-human-centered works' visibility and discourage creative diversity.

(4) Moreover, expert-driven annotation, while ensuring consistency, may differ from public aesthetic preferences, leading to a gap between technical evaluation and social perception as the sole reference.

To address these issues, we recommend the following responsible-use guidelines:

**(1) Clarify applicability:** Users should recognize that the proposed aesthetic standards and model judgments are culturally contextual and should not be treated as universal criteria.

**(2) Mitigate bias**: Future work should involve annotators from diverse backgrounds, integrate feedback from general audiences, and use multi-source data to reduce single-perspective bias.

## 6 Conclusions

VADB, the largest and most comprehensively annotated video aesthetics dataset to date, derives core value from multi-dimensional annotations: scores, attributes, comments, and tags. Specifically, it provides fine-grained overall and attribute-specific aesthetic scores, enriched with detailed language comments and objective technical tags, achieving the first synergistic annotation of quantitative aesthetic analysis and semantic descriptions for videos. Through systematic scoring criteria and rigorous quality control by a professional annotation team, VADB establishes a reliable data foundation for video aesthetics research, addressing the long-standing absence of standardized datasets in this field.

Furthermore, the proposed VADB-Net validates the effectiveness and superiority of the CLIP architecture in video aesthetic quality assessment tasks, which effectively combines general visual representations with specialized aesthetic knowledge, significantly enhancing the accuracy of aesthetic scoring. VADB-Net not only outperforms existing video quality assessment models but also offers a pre-trained video encoder that supports flexible adaptation to other aesthetic assessment tasks, providing a novel technical pathway for computational video aesthetics.

Future work will expand the dataset's video categories and volume, incorporating public aesthetics and cross-cultural variations to enhance diversity and universality. Additionally, we aim to upgrade the scoring model for improved aesthetic evaluation in complex scenarios. Furthermore, we plan to extend VADB-Net to emerging applications (e.g., AIGC video generation quality assessment, industrialized film production) to foster practical adoption of video aesthetics research.

## Acknowledgments

We thank the ACs, reviewers, and annotators, as well as the experts across industry, academia, and research institutions for their support and discussions. We would like to thank the Ling team at Ant Group for their profound expertise in hybrid architectures and multimodal evaluation, which has provided the study with critical technical insights and constructive feedback. This work is partially supported by the National Natural Science Foundation of China (62476013) and the Opening Project of the State Key Laboratory of General Artificial Intelligence, BIGAI/Peking University, Beijing, China (Project No. SKLAGI2025OP01).

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

# A  Open Source Materials Summary

**VADB Scoring Criteria:** *Scoring Criteria and Example Videos of VADB*

**Tag Annotation Criteria:** *Tag Annotation Criteria and Example Videos of VADB*

**Dataset:** *BestiVictoryLab/VADB* (hosted on Hugging Face)

**Code and Models:** *BestiVictory/VADB* (hosted on GitHub)

# B  Dataset Comparison Table

In the table below, we compare the VADB dataset with existing video aesthetic datasets. It can be seen from the comparison that VADB has the largest number of videos and the most comprehensive annotation perspectives. It also indicates that video aesthetic datasets are quite scarce and urgently need to be expanded.

Table 6: Comparison of different video aesthetic datasets

| Dataset Name | Number of Videos | Annotation Type | Characteristics |
|---|---|---|---|
| Telefonica dataset [Bylinskii et al., 2015] | 160 | Ratings | An early video aesthetic dataset, annotated based on YouTube videos. |
| Niu et al.'s dataset [Niu and Liu, 2012] | 2,000 | Binary classification | Binary classification into professional or amateur videos. |
| VAQ700 dataset [Tzelepis et al., 2016] | 700 | Ratings | Targeting daily life videos, with ratings from multiple annotators. |
| AVAQ6000 [Kuang et al., 2019] | 6,000 | Binary classification | Binary classification of professional or amateur for drone videos. |
| DIVIDE-3k [Wu et al., 2023] | 3,590 | Ratings | Aesthetic rating is only part of the overall rating. |
| VADB (ours) | 10,490 | 11 types of ratings, comments, and objective tags | Currently the largest in scale, with rich annotation dimensions, annotated by professionals. |

# C  Selection of Video Aesthetic Attributes

To construct a broadly applicable dataset for video aesthetic scoring and commentary, this study meticulously selects and designs both universal and specialized aesthetic attributes based on film and media aesthetics theory [Wu, 2024, Matbouly, 2022, Petrogianni et al., 2022, Arijon, 2013]. These attributes comprehensively evaluate the visual expressiveness of videos, accommodating the diverse sources and varying quality levels of the dataset while providing a unified standard for assessing different video types.

## C.1  Universal Aesthetic Attributes

The composition of the frame serves as the foundation for effective visual communication. Whether in professionally produced documentaries, narrative films, or casually recorded videos, a well-structured layout prevents cluttered elements and subject imbalance, ensuring comparability in visual order across diverse video sources. Shot scale, a critical element of visual storytelling, must align with narrative intent, from panoramic shots in news footage to close-ups in variety shows. This attribute is evaluated based on content appropriateness, mitigating biases arising from differing creative objectives.

Lighting and tonality are pivotal for visual presentation and atmosphere creation. From AI-generated virtual scenes to real-world documentaries, adequate and purposeful lighting ensures subject clarity, while tonality conveys the intended mood. Whether aiming for objective realism in news or artistic expression in films, tonality must remain logically consistent with the content. Color directly influences emotional perception, with evaluations distinguishing between technically accurate natural reproduction and stylistically intentional artistic coloring, ensuring comparability in color appropriateness. Depth of field, as a tool for guiding visual focus, varies across video types but must consistently maintain clear focal intent, avoiding issues like blurred subjects or distracting backgrounds.

## C.2  Specialized Attributes for Character-Driven Videos

For videos featuring human subjects, four specialized attributes—expression, movement, costume, and makeup—further refine the evaluation. Facial expression, central to conveying emotion, is assessed based on the efficiency of emotional delivery, whether capturing subtle micro-expressions or portraying dramatic performances in films. Movement, a key narrative vehicle, must adhere to physical plausibility and expressive clarity, whether documenting authentic actions in news or choreographed sequences in films. Costume and makeup serve as visual indicators of character identity and scene appropriateness, evaluated for contextual fit and visual coherence, from culturally accurate attire in documentaries to role-specific costumes and special-effects makeup in narrative films.

## C.3  Evaluation Framework

This attribute system employs a layered evaluation framework. At the technical level, it establishes a baseline quality threshold to ensure comparability across videos of varying quality. At the creative level, it preserves artistic freedom while ensuring logical consistency. By focusing on visual outcomes, the framework eliminates dependency on filming equipment, enabling compatibility with both real-world recordings and virtual creations. This system not only facilitates direct comparisons across diverse video sources and highlights aesthetic characteristics of specific video types but also provides structured labels for training multidimensional aesthetic evaluation models, enhancing the dataset's utility for academic research and industrial applications.

# D   Logic Behind Establishing Video Aesthetic Standards

## D.1  Initial Considerations

In the initial phase of constructing the annotation framework, several principles were established to ensure the objectivity and validity of the evaluation process. Objectivity was prioritized, requiring annotators to adopt a general audience perspective and minimize subjective biases. Flexibility was also emphasized, particularly for avant-garde or experimental videos, allowing slight adjustments to the established standards to assess their uniqueness, accompanied by textual explanations. To facilitate comprehensive evaluation, the relationship between overall scores and attribute-specific scores was clarified: annotators were required to maintain independence between overall and attribute scores. A video with an excellent overall visual effect could receive a high overall score despite lower scores in certain attributes, while a video with poor overall impact could still achieve high attribute scores for standout elements. These principles provided a foundation for subsequent standard development.

## D.2  Establishment of Video Content Categories

Recognizing that different video types emphasize distinct aesthetic qualities, videos were categorized into four types: "character," "landscape," "architecture," and "food." This classification was based on the prevalent themes in mainstream video content and the distinct visual elements of each category. For instance, "character" videos focus on character expressions, movements, and costume design, while "landscape" videos emphasize visual impact, atmosphere, and composition. Annotation dimensions were tailored to reflect the unique characteristics of each category.

### D.3 Development of Scoring System and Attribute Dimensions

For each video category, an evaluation structure was designed, comprising an overall score with comments and multiple attribute-specific scores.

Overall Scoring System: A 10-point scale (1–10) was adopted, with each score level accompanied by distinct evaluation criteria and benchmarked by "reference videos." These ranged from "severe deficiencies across all aspects" (1 point) to "flawless masterpiece" (10 points), covering the full spectrum of video quality. The overall score provided annotators with clear reference points to reduce subjective variability, and annotators were required to provide a brief textual evaluation of the video's overall aesthetic quality. Attribute Dimensions: Specific attributes were defined for each category, with tailored criteria to reflect their unique aesthetic priorities.

### D.4 Application of Reference Videos

To enhance annotation consistency and accuracy, each score level and attribute tier was paired with corresponding "reference videos" representing the respective quality standard. Annotators could refer to these videos before and during the annotation process to calibrate their judgments, ensuring quality control in large-scale annotation tasks.

## E VADB-Net

### E.1 Pre-training Stage

#### E.1.1 Model Architecture

Refer to CLIP4CLIP[Luo et al., 2021], the Pre-training Stage model extends the CLIP framework (ViT-B/32) for video-text retrieval, comprising a video encoder, dual text encoders, and a dynamic fusion module. The video encoder leverages CLIP's 12-layer visual Transformer (`vision_layers=12`), employing 3D convolution initialization (`linear_patch="3d"`) to extend 2D convolutional kernels for capturing spatiotemporal features. It processes 12 video frames (`max_frames=12`, resolution 224×224) and generates a 512-dimensional video representation via mean pooling (`sim_header="meanP"`).

The dual text encoder architecture includes a primary text encoder (`clip`) and a tag encoder (`clip_tag`), both utilizing CLIP's 12-layer Transformer (`transformer_layers=12`) to process natural language captions and aesthetic tags (e.g., "symmetric composition"), respectively, yielding 512-dimensional features. The tag encoder shares the visual encoder's weights but maintains independent text encoder parameters.

The dynamic fusion module (`DynamicFusion`) integrates the two text features using an attention mechanism, computing a weight $\alpha$ via a two-layer fully connected network (`nn.Linear` $\rightarrow$ `nn.Tanh` $\rightarrow$ `nn.Linear`) with initial biases `text_bias=0.7` and `tag_bias=0.3`. The fused feature is defined as:

$$f_{\text{fused}} = \alpha \cdot f_{\text{proj}}^{\text{text}} + (1 - \alpha) \cdot f_{\text{proj}}^{\text{tag}} \tag{1}$$

A cross-modal encoder (`CrossModel`, 4 layers, `cross_num_hidden_layers=4`) further processes the fused text and video features to produce the final representation. The code implementation is publicly available; see the project repository for details.

#### E.1.2 Data Loading and Preprocessing

Data loading and preprocessing are implemented via the `VADB_TrainDataLoader` class, supporting the VADB dataset (226,940 video-text-tag samples). The dataset metadata include a video ID list, video-text pairs with associated tags, and video files . When `unfold_sentences=True`, the `sentences_dict` structure unfolds each video-text-tag triplet into an independent sample, handling missing tags by padding with empty strings. When `unfold_sentences=False`, the `sentences` structure aggregates captions and tags by video ID, randomly selecting sample pairs. In this article, unfold_sentences is set to True by default.

Text preprocessing employs the `ClipTokenizer` for tokenization, incorporating special tokens (`<|startoftext|>`, `<|endoftext|>`, `[PAD]`), with a fixed sequence length of 32. Video prepro-

cessing uses the `RawVideoExtractor` to extract 12 frames at 1 fps (`feature_framerate=1`) with uniform sampling (`slice_framepos=2`), producing tensors of shape [`batch_size, 12, 1, 3, 224, 224`] alongside corresponding video masks. Multi-threaded loading (`num_thread_reader=8`) enhances data retrieval efficiency. Details of the data processing pipeline are available in the open-source code repository.

### E.1.3 Training Procedure and Optimization

The training procedure is implemented in the `train_epoch` function, processing each batch comprising primary text inputs (`input_ids`, `attention_mask`, `segment_ids`), tag text inputs (`input_ids_tag`, `attention_mask_tag`, `segment_ids_tag`), and video data (`video`, `video_mask`). The model's forward pass computes the contrastive loss using the `CrossEn` function, optimizing text-video feature alignment through bidirectional cross-entropy losses (`sim_loss1` for text-to-video and `sim_loss2` for video-to-text). The `BertAdam` optimizer is employed, with parameters grouped into CLIP and non-CLIP modules. The learning rate for CLIP parameters is scaled by a coefficient (`coef_lr=1e-3`), with an initial learning rate of 0.0001, a 10% warmup proportion, a decay rate of 0.9, and a weight decay of 0.2. Gradient clipping (`max_grad_norm=1.0`) ensures training stability. Distributed training leverages `torch.distributed` and `DistributedDataParallel`, utilizing four NVIDIA H20 GPUs with a batch size of 64 over two epochs. Model and optimizer states are saved per epoch via the `save_model` function, with support for resuming training from checkpoints . Training logs are recorded every 50 steps. Implementation details are available in the open-source code repository.

### E.1.4 Loss Function and Similarity Computation

The similarity computation is implemented in the `_loose_similarity` method of the `CLIP4Clip` class. It calculates the cosine similarity between fused text features and video features via matrix multiplication, scaled by a learnable temperature parameter (`logit_scale`, initialized as $\ln(100)$). Video features are processed through mean pooling (`_mean_pooling_for_similarity_visual`), and both text and video features are L2-normalized to ensure unit length. In distributed training, the `AllGather` operation synchronizes features across GPUs to construct a complete similarity matrix. The loss function employs `CrossEn`, computing bidirectional cross-entropy losses for text-to-video and video-to-text alignments, which are averaged to form the total loss, ensuring cross-modal feature space alignment. The parameters `margin=0.1` and `hard_negative_rate=0.5` further optimize the selection of negative samples. The similarity computation adopts the `meanP` strategy.

### E.1.5 Experimental Setup

The experiments were conducted on the VADB dataset, comprising 226,940 video-text-tag samples. Video inputs consist of 12 frames sampled at 1 frame per second with a resolution of $224 \times 224$. Text inputs are tokenized to a maximum sequence length of 32 tokens. The training configuration includes a batch size of 64, an initial learning rate of 0.0001, a warmup proportion of 10%, a learning rate decay of 0.9, and training for 2 epochs. The model architecture is parameterized with 12 layers for both the visual and text Transformers, 4 layers for the cross-modal encoder, and an embedding dimension of 512. Training was performed on 4 NVIDIA H20 GPUs with distributed training enabled. Experimental results and model weights are publicly available, and readers can reproduce the experiments via the project repository.

### E.2 Fine-tuning Stage

The aesthetic score regression model comprises a pretrained Visual Encoder and an additional multi-layer perceptron (MLP), denoted as `AestheticPredictor`, designed to predict continuous aesthetic scores (ranging from 0 to 10) from video content. The `AestheticPredictor` is a three-layer MLP with two hidden layers (dimensions 512 and 256, and a single output layer (`output_dim=1`), utilizing ReLU activation functions to map the 512-dimensional video features to a single aesthetic score. The Visual Encoder's parameters are frozen during training (`param.requires_grad = False`), with only the MLP parameters optimized to enhance efficiency. The model is initialized with pretrained weights, ensuring robust feature extraction. The implementation is publicly available in the project repository.

The model is trained with a batch size of 128 for training and 32 for validation, leveraging multi-threaded data loading (`num_thread_reader=4`) for efficiency. The loss function employs mean squared error (MSELoss), computing the squared difference between predicted and ground-truth scores, defined as:

$$\text{loss} = \frac{1}{N} \sum_{i=1}^{N} (\text{predicted\_score}_i - \text{score}_i)^2,$$

where $N$ is the batch size. During the validation phase, the average test loss is calculated, and the model with the lowest validation loss is saved. Training is conducted on a single GPU using the Adam optimizer with a learning rate of $1 \times 10^{-4}$. Training progress is logged every 50 steps, with detailed implementation available in the open-source codebase.

**Overall Score Model**: The design objective is to predict the overall aesthetic quality score of a video, reflecting its comprehensive aesthetic value. The MLP network is defined in the AestheticPredictor class, structured as a three-layer MLP comprising two hidden layers and an output layer, which maps to a single aesthetic score.

**Attribute Score Model**: The design objective is to simultaneously predict multiple aesthetic attribute scores of a video, with each attribute outputting a scalar score, framing the task as a multi-output regression problem. The MLP network consists of six parallel AestheticPredictor branches, each sharing the same architecture as the single overall score MLP. Each branch independently predicts the score of one aesthetic attribute, with no parameter sharing between branches. This increases the model's parameter count but enhances flexibility in multi-attribute modeling.

# F   Cross-dataset Experiments

In the cross-dataset comparison experiment, we trained four models (SimpleVQA, Fast-VQA, ModularBVQA, and VADB-net) on the VADB dataset and tested these models on the DIVIDE-3k dataset. The obtained results are shown in Table 7.

Table 7: Cross-dataset comparison results (trained on VADB, tested on DIVIDE-3k). Bold values indicate the best performance for each metric.

| Evaluation Metrics | SimpleVQA | Fast-VQA | ModularBVQA | VADB-net |
|:---:|:---:|:---:|:---:|:---:|
| SRCC↑ | 0.35 | 0.30 | **0.43** | 0.42 |
| PLCC↑ | 0.38 | 0.29 | 0.44 | **0.46** |
| KRCC↑ | 0.24 | 0.20 | **0.29** | **0.29** |
| RMSE↓ | 0.54 | 0.69 | **0.52** | 3.74 |

The cross-dataset performance of all models, though with varying degrees, remains less satisfactory. This can primarily be attributed to the significant discrepancies between VADB and DIVIDE-3k: on one hand, there exists a considerable gap in video features between the two datasets, and on the other hand, their aesthetic scoring criteria differ markedly. Such differences lead to a mismatch where the patterns learned by models from VADB fail to align with the evaluation logic inherent in DIVIDE-3k, thereby hindering the models' generalization ability.

However, it should be noted that currently, DIVIDE-3k is the only open-source dataset available for testing video aesthetic scoring with ground-truth annotations. Due to the lack of other datasets with reliable aesthetic truth values, our cross-dataset comparison could only be conducted on DIVIDE-3k. Therefore, it would be inappropriate to conclude that the annotations of VADB have poor generality or that the model trained on VADB has weak transferability based solely on these results. A more comprehensive assessment would require further validation with additional diverse datasets in future studies.

# G   Statistical Significance

The Complete Statistical Significance Analysis is shown in Table 8.

Table 8: Complete Statistical Significance Analysis

| Score Type | Index | Value | P-value | 95% Confidence Interval | Error Bars (Lower/Upper) |
|---|---|---|---|---|---|
| Overall Score | MSE | 0.2941 | - | [0.2718, 0.3186] | (0.0224, 0.0244) |
| | SROCC | 0.9299 | <0.001 | [0.9232, 0.9353] | (0.0067, 0.0054) |
| | PLCC | 0.9305 | <0.001 | [0.9242, 0.9372] | (0.0063, 0.0067) |
| | KRCC | 0.7704 | <0.001 | [0.7602, 0.7811] | (0.0101, 0.0107) |
| | ACC | 0.9180 | <0.001 | [0.9061, 0.9295] | (0.0119, 0.0114) |
| Composition | MSE | 0.2697 | - | [0.2494, 0.2935] | (0.0203, 0.0238) |
| | SROCC | 0.9292 | <0.001 | [0.9230, 0.9344] | (0.0062, 0.0051) |
| | PLCC | 0.9421 | <0.001 | [0.9372, 0.9469] | (0.0050, 0.0047) |
| | KRCC | 0.7661 | <0.001 | [0.7558, 0.7755] | (0.0103, 0.0094) |
| | ACC | 0.9518 | <0.001 | [0.9428, 0.9619] | (0.0091, 0.0100) |
| Shot Size | MSE | 0.2579 | - | [0.2384, 0.2786] | (0.0196, 0.0207) |
| | SROCC | 0.9287 | <0.001 | [0.9227, 0.9337] | (0.0060, 0.0050) |
| | PLCC | 0.9404 | <0.001 | [0.9352, 0.9452] | (0.0052, 0.0048) |
| | KRCC | 0.7638 | <0.001 | [0.7547, 0.7730] | (0.0091, 0.0092) |
| | ACC | 0.9428 | <0.001 | [0.9323, 0.9523] | (0.0105, 0.0095) |
| Lighting | MSE | 0.3519 | - | [0.3209, 0.3846] | (0.0309, 0.0328) |
| | SROCC | 0.9258 | <0.001 | [0.9186, 0.9318] | (0.0072, 0.0060) |
| | PLCC | 0.9203 | <0.001 | [0.9126, 0.9276] | (0.0077, 0.0073) |
| | KRCC | 0.7630 | <0.001 | [0.7529, 0.7725] | (0.0101, 0.0095) |
| | ACC | 0.9103 | <0.001 | [0.8979, 0.9223] | (0.0124, 0.0119) |
| Visual Tone | MSE | 0.3785 | - | [0.3440, 0.4162] | (0.0345, 0.0378) |
| | SROCC | 0.9186 | <0.001 | [0.9102, 0.9257] | (0.0084, 0.0071) |
| | PLCC | 0.9167 | <0.001 | [0.9080, 0.9246] | (0.0087, 0.0079) |
| | KRCC | 0.7540 | <0.001 | [0.7421, 0.7653] | (0.0119, 0.0113) |
| | ACC | 0.9199 | <0.001 | [0.9084, 0.9309] | (0.0114, 0.0110) |
| Color | MSE | 0.5509 | - | [0.4989, 0.6106] | (0.0520, 0.0596) |
| | SROCC | 0.8902 | <0.001 | [0.8790, 0.8995] | (0.0112, 0.0092) |
| | PLCC | 0.8844 | <0.001 | [0.8719, 0.8954] | (0.0125, 0.0110) |
| | KRCC | 0.7151 | <0.001 | [0.7007, 0.7279] | (0.0143, 0.0128) |
| | ACC | 0.9108 | <0.001 | [0.8989, 0.9227] | (0.0119, 0.0119) |
| Depth of Field | MSE | 0.4348 | - | [0.3915, 0.4814] | (0.0433, 0.0466) |
| | SROCC | 0.9005 | <0.001 | [0.8903, 0.9090] | (0.0102, 0.0085) |
| | PLCC | 0.8972 | <0.001 | [0.8857, 0.9072] | (0.0115, 0.0099) |
| | KRCC | 0.7284 | <0.001 | [0.7151, 0.7405] | (0.0133, 0.0120) |
| | ACC | 0.9065 | <0.001 | [0.8941, 0.9194] | (0.0124, 0.0129) |
| Expression | MSE | 0.601 | - | [0.533, 0.681] | (0.068, 0.080) |
| | SROCC | 0.901 | <0.001 | [0.886, 0.911] | (0.014, 0.011) |
| | PLCC | 0.890 | <0.001 | [0.876, 0.902] | (0.014, 0.012) |
| | KRCC | 0.731 | <0.001 | [0.716, 0.747] | (0.015, 0.015) |
| | ACC | 0.927 | <0.001 | [0.915, 0.939] | (0.012, 0.012) |
| Movement | MSE | 0.479 | - | [0.422, 0.542] | (0.056, 0.063) |
| | SROCC | 0.913 | <0.001 | [0.902, 0.922] | (0.011, 0.009) |
| | PLCC | 0.894 | <0.001 | [0.882, 0.906] | (0.013, 0.012) |
| | KRCC | 0.745 | <0.001 | [0.733, 0.758] | (0.012, 0.013) |
| | ACC | 0.903 | <0.001 | [0.887, 0.916] | (0.015, 0.013) |
| Costume | MSE | 0.429 | - | [0.382, 0.486] | (0.047, 0.057) |
| | SROCC | 0.911 | <0.001 | [0.898, 0.921] | (0.013, 0.010) |
| | PLCC | 0.907 | <0.001 | [0.897, 0.917] | (0.010, 0.010) |
| | KRCC | 0.749 | <0.001 | [0.734, 0.763] | (0.016, 0.014) |
| | ACC | 0.921 | <0.001 | [0.907, 0.934] | (0.014, 0.014) |
| Makeup | MSE | 1.008 | - | [0.872, 1.159] | (0.137, 0.151) |
| | SROCC | 0.871 | <0.001 | [0.852, 0.887] | (0.020, 0.016) |
| | PLCC | 0.839 | <0.001 | [0.816, 0.859] | (0.023, 0.020) |
| | KRCC | 0.700 | <0.001 | [0.684, 0.717] | (0.016, 0.016) |
| | ACC | 0.894 | <0.001 | [0.879, 0.908] | (0.015, 0.014) |

# H    Test Sample

Below are five model test examples for videos, where G_T represents the ground-truth scores of the videos and Predicted indicates the predicted values from the model.

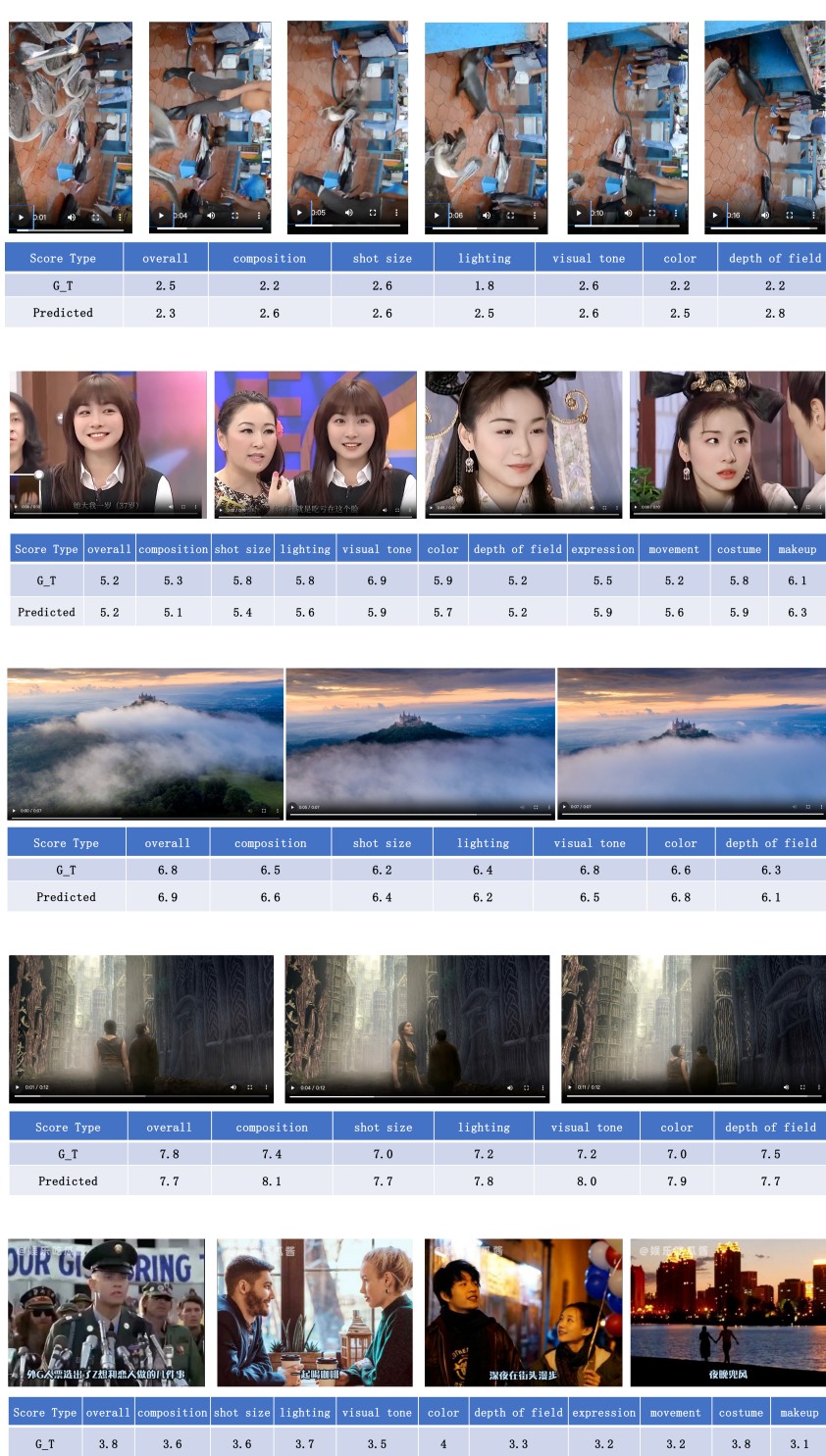

| Score Type | overall | composition | shot size | lighting | visual tone | color | depth of field |
|---|---|---|---|---|---|---|---|
| G_T | 2.5 | 2.2 | 2.6 | 1.8 | 2.6 | 2.2 | 2.2 |
| Predicted | 2.3 | 2.6 | 2.6 | 2.5 | 2.6 | 2.5 | 2.8 |

| Score Type | overall | composition | shot size | lighting | visual tone | color | depth of field | expression | movement | costume | makeup |
|---|---|---|---|---|---|---|---|---|---|---|---|
| G_T | 5.2 | 5.3 | 5.8 | 5.8 | 6.9 | 5.9 | 5.2 | 5.5 | 5.2 | 5.8 | 6.1 |
| Predicted | 5.2 | 5.1 | 5.4 | 5.6 | 5.9 | 5.7 | 5.2 | 5.9 | 5.6 | 5.9 | 6.3 |

| Score Type | overall | composition | shot size | lighting | visual tone | color | depth of field |
|---|---|---|---|---|---|---|---|
| G_T | 6.8 | 6.5 | 6.2 | 6.4 | 6.8 | 6.6 | 6.3 |
| Predicted | 6.9 | 6.6 | 6.4 | 6.2 | 6.5 | 6.8 | 6.1 |

| Score Type | overall | composition | shot size | lighting | visual tone | color | depth of field |
|---|---|---|---|---|---|---|---|
| G_T | 7.8 | 7.4 | 7.0 | 7.2 | 7.2 | 7.0 | 7.5 |
| Predicted | 7.7 | 8.1 | 7.7 | 7.8 | 8.0 | 7.9 | 7.7 |

| Score Type | overall | composition | shot size | lighting | visual tone | color | depth of field | expression | movement | costume | makeup |
|---|---|---|---|---|---|---|---|---|---|---|---|
| G_T | 3.8 | 3.6 | 3.6 | 3.7 | 3.5 | 4 | 3.3 | 3.2 | 3.2 | 3.8 | 3.1 |
| Predicted | 3.9 | 3.7 | 3.4 | 3.7 | 3.6 | 3.6 | 3.5 | 3.4 | 3.2 | 3.8 | 3.1 |

