# OpenReview forum: "VADB: A Large-Scale Video Aesthetic Database with Professional and Multi-Dimensional Annotations"
_NeurIPS.cc/2025/Datasets_and_Benchmarks_Track — NeurIPS 2025 Datasets and Benchmarks Track poster_

### Official Review · Reviewer_s4fd · 2025-06-21

**Rating:** 5
**Confidence:** 4

**Summary:**

The paper presents VADB, a large-scale Video Aesthetic Database consisting of 10,490 videos annotated by 37 professionals across multiple aesthetic dimensions. The authors introduce VADB-Net, a dual-modal pre-training framework that utilizes a two-stage training strategy, showing superior performance in aesthetic scoring tasks compared to existing models. The dataset includes fine-grained aesthetic scores, rich language comments, and objective tags, addressing the lack of standardized datasets in video aesthetics research. Experimental results demonstrate VADB-Net's effectiveness in aesthetic quality assessment.

**Dataset Code Accessibility:**

Yes

**Ethical Considerations:**

No, there are no or only very minor ethics concerns

**Final Justification:**

Thanks to the authors' rebuttal, my concerns have been addressed. I will keep my accept rating.

**Limitations Weaknesses:**

- The paper presents an inconsistency between the definition of aesthetic attributes and the types of videos included in the dataset. For instance, Table 1 lists character-specific attributes, such as expression, which are applicable primarily to character videos. However, the dataset encompasses four distinct video types: character, landscape, architecture, and food. This raises questions about how aesthetic attributes relevant to character videos can be effectively applied to landscape, architecture, and food videos. Furthermore, this inconsistency may limit the scope of the video aesthetic assessment, potentially affecting the generalizability of the findings across different videos.

- The definition of the aesthetic score remains vague and lacks clarity. While Fig. 2 provides some guidelines for labeling aesthetic scores, the reviewer finds it difficult to grasp the specific criteria used for scoring. For instance, the significance of a score of 4.5 for a video is not clearly explained.

- The expression attribute is indeed multifaceted, encompassing various classes and intensities that significantly contribute to a video's aesthetic quality. However, the implications of the presence or absence of this expression attribute are not clearly defined. In the context of video, expressions can vary over time, leading to different aesthetic interpretations at different moments within the same video. It would be beneficial for the authors to elaborate on how the expression attribute is assessed and its impact on the overall aesthetic evaluation.

[A] Aesthetics and Emotions in Images, SPM, 2011.

[B] Weakly Supervised Video Emotion Detection and Prediction via Cross-Modal Temporal Erasing Network, CVPR, 2023.

- Writing. Fig. 2 and 3 are not cited in the paper.

**Strengths Contributions:**

+ The dataset is large, dense-annotated and will contribute to the community.
+ The paper is well-written and well-structured.
+ The dataset construction is solid.

---

> ### Author Rebuttal · Authors · 2025-07-29
>
> Thank you for your valuable review comments and recognition of our work. In response to the limitations and weaknesses you have pointed out, our replies are as follows:
>
> 1.Person-specific attributes (e.g., "expression") cannot be effectively applied to landscape, architecture, and food videos.
>
> Reply: You are correct. Person-specific attributes (include expression, movement, costume, and makeup) are not applicable to landscape, architecture, or food videos and are only annotated for person-related videos. We will clarify this point in the paper to avoid misunderstandings among readers.
>
> 2.The definition of aesthetic scoring is relatively vague, and reviewers cannot easily understand the specific basis for scoring solely based on Figure 2. For example, the significance of a score of 4.5 for a video is not clearly explained.
>
> Reply: We have specifically developed aesthetic scoring criteria for different video categories, attributes, and score ranges, with 3 video examples provided for each score range to ensure annotators can clearly follow the standards. However, this standard document is too extensive to be fully presented in the main text. If this paper is fortunate enough to be accepted, we will add a link to the web version of the scoring criteria and tag selection standards in the final submitted version.
>
> In the scoring criteria and actual annotation process, scores are integers within the range of 1-10. Since each video is annotated by 10 people, the final score is calculated as an average, which may result in decimal values. For instance, a video scoring 4.5 indicates that its aesthetic quality falls between 4 and 5 points.
>
> 3.The impact of the presence or absence of the "expression" attribute has not been clearly defined. Changes in facial expressions can lead to different aesthetic interpretations of the same video at different moments. It would be beneficial for the authors to elaborate on how the “expression” attribute is assessed and its impact on the overall aesthetic evaluation.
>
> Reply: As you noted, the "expression" attribute involves multiple dimensions, including different classes and intensities, and has a significant impact on the aesthetic quality of videos. Due to its variability, this attribute is more difficult to measure than conventional aesthetic attributes such as "composition" and "visual tone". We will refer to the references you provided and clarify the importance of the facial expression attribute in the final version of the paper, while elaborating on its evaluation criteria and impact on the overall aesthetic assessment.
>
> 4.Writing issue: Figures 2 and 3 are not cited in the paper.
>
> Reply: Thank you for your careful reading and inspection. We will add citations to Figures 2 and 3 in the final version of the paper and conduct a thorough check of other parts of the article to strictly ensure its readability and rigor.

---

> > ### Comment · Reviewer_s4fd · 2025-08-04
> > **Official Comments**
> >
> > Thanks to the authors' rebuttal, my concerns have been addressed. I will keep my accept rating.

---

### Official Review · Reviewer_ZFYz · 2025-07-01

**Rating:** 4
**Confidence:** 4

**Summary:**

In this work, the authors present VADB, a significant large-scale video aesthetic database containing 10,490 videos, annotated comprehensively by a team of at least 13 professional annotators per video. This dataset features diverse annotations across multiple aesthetic dimensions, including overall and attribute-specific scores, rich descriptive comments, and precise technical tags. Furthermore, the authors propose VADB-Net, a novel dual-modal pre-training framework based on the CLIP architecture, which employs a two-stage training strategy. Their experimental results show that VADB-Net surpasses existing models in the VADB benchmark tasks and demonstrates considerable effectiveness for downstream video aesthetic assessments.

**Additional Feedback:**

I will be happy to give it an "accept" if my comments on the weaknesses are addressed.

**Dataset Code Accessibility:**

Yes

**Ethical Considerations:**

No, there are no or only very minor ethics concerns

**Final Justification:**

The potential impact of the paper is valuable. However, the authors skipped several key experiments (as other reviewers also raised) and submitted them in the rebuttal. Hence, I suggest the paper go through another round of revision before submission because it is very important to demonstrate the impact of a dataset paper with rigorous experiments. Hence, I am keeping my score as it is.

**Limitations Weaknesses:**

One notable limitation is the lack of clarity regarding the granularity of annotations. The authors do not address how aesthetic qualities might shift within individual clips. Without an explicit metric or discussion on intra-clip aesthetic variation, the dataset leaves unresolved how rapid stylistic or compositional changes impact the reliability of aesthetic scoring. Further, it is unclear how annotators were instructed to manage videos with frequent or abrupt transitions in aesthetic qualities.

Although video aesthetics differ fundamentally from image aesthetics due to temporal dimensions, the paper omits a detailed comparison or contrast with established image aesthetic datasets. Including such a discussion would clarify the novelty of VADB and better inform potential users about opportunities and challenges when transferring methodologies from image aesthetic evaluation to video aesthetics.

**Strengths Contributions:**

1. The scale of the VADB dataset, consisting of over 10,000 annotated videos, is a substantial achievement that addresses the longstanding scarcity of comprehensive, standardized, and high-quality datasets for video aesthetics research. Each video is evaluated meticulously by at least 13 annotators, resulting in robust and reliable data suitable for developing advanced computational aesthetics models.

2. The rich and multi-dimensional annotations in VADB, which include overall aesthetic scores, ratings on ten specific aesthetic attributes, detailed language comments, and seven objective technical tags, make the dataset particularly versatile. These annotations facilitate diverse research applications, ranging from genre-specific analyses to multimodal model pre-training, significantly broadening the potential impact of this dataset on video aesthetics studies.

3. The authors have developed a clearly structured scoring framework complemented by textual descriptions and illustrative exemplar videos, thereby ensuring transparency and interpretability of the scoring. This approach considerably strengthens the reproducibility and reliability of annotations, which is critical for effective benchmarking.

4. Annotations were systematically conducted by 37 trained professionals from the Beijing Film Academy, guided by stringent recruitment criteria, comprehensive training protocols, and ongoing quality control measures. This meticulous process significantly enhances the credibility, consistency, and reliability of the dataset, ensuring high-quality annotations throughout the database.

---

> ### Author Rebuttal · Authors · 2025-07-29
>
> Thank you for your valuable review comments and recognition of our work. In response to the limitations and weaknesses you have pointed out, our replies are as follows:
>
> 1.The granularity of annotations is not clear enough. Aesthetic traits within a single video clip may change, but there is a lack of clear annotation standards or relevant discussions on intra-clip aesthetic differences, leaving annotators without guiding principles when processing similar videos.
>
> Reply: It is true that aesthetic characteristics within the same video clip may vary. However, overly fine-grained annotation would require the establishment of an extremely large-scale standard system, which would pose significant challenges to standard formulation, training of annotators, and the actual annotation process. Therefore, we have entrusted this judgment to professional annotators. Although there may be differences in the handling methods among different annotators, we trust the professionalism of our annotation team. Moreover, the final annotation results integrate the "comprehensive judgments" of all participants, so we remain confident in the reliability of the annotation outcomes. In future extended versions of VADB, we will not only consider the overall aesthetic characteristics of different types of videos but also take into account the aesthetic traits of different scenes within the same video at a finer granularity, making the aesthetic annotations of the dataset more rigorous, detailed, and reliable.
>
> 2.No comparison with image aesthetic datasets was conducted.
>
> Reply: Thank you for this suggestion. Comparing with image aesthetic datasets would indeed better highlight the scarcity and importance of video aesthetic datasets, as well as clarify the innovativeness of the VADB dataset. If this paper is fortunate enough to be accepted, we will add a comparison between VADB and image aesthetic datasets in the final submitted version.

---

### Official Review · Reviewer_Bkpp · 2025-07-02

**Rating:** 5
**Confidence:** 3

**Summary:**

1.This study introduced VADB, the largest video aesthetic database with 10,490 diverse videos annotated by 37 professionals across multiple aesthetic dimensions, including overall and attribute-specific aesthetic scores, rich language comments and objective tags.
2. The authors also proposed VADB-Net, a dual-modal pre-training framework with a two-stage training strategy, which outperforms existing video quality assessment models in scoring tasks and supports downstream video aesthetic assessment tasks.

**Dataset Code Accessibility:**

Yes

**Ethical Considerations:**

Yes, there are ethics concerns that require attention by the authors

**Ethics Flags:**

["Data privacy, copyright, and consent", "Discrimination, bias, and fairness", "Environmental impact"]

**Final Justification:**

Thanks the authors' rebuttal, my concerns have been addressed. I will keep my accept rating.

**Limitations Weaknesses:**

1.There are only four category videos. There are still too few types.
2.Different video types emphasize distinct aesthetic qualities. While it brings benefits, the inability to make a uniform evaluation without discrimination is a problem.

**Strengths Contributions:**

1. A set of video aesthetic annotation criteria. A detailed framework developed by a team of film and television professionals, outlining the scoring criteria for an overall aesthetic score, 10 specific attribute scores, and selection guidelines for 34 technical tags.
2.The largest video aesthetic database with 10,490 diverse videos annotated by 37 professionals across multiple aesthetic dimensions, including overall and attribute-specific aesthetic scores, rich language comments and objective tags
3. The VADB-Net, a dual-modal pre-training framework with a two-stage training strategy, delivering superior scoring performance.

---

> ### Author Rebuttal · Authors · 2025-07-29
>
> Thank you for your valuable review comments and recognition of our work. In response to the limitations and weaknesses you have pointed out, our replies are as follows:
>
> 1.Only four category videos are included, and the number of types is too small.
>
> Reply: The reason why VADB only includes four types of videos (portrait, food, architecture, and landscape) is that we need to formulate annotation standards for each type. These standards cover scoring criteria for overall scores and individual attribute scores, as well as annotation rules for a large number of tags, which involves substantial workload in standard formulation. Therefore, we chose four of the most common video types to build the dataset. However, to enrich the diversity of video categories in the dataset, we have further subdivided the portrait category (which has the largest number of videos) into sub-types: film and television, variety shows, news, and random shots. We plan to continue expanding the video categories of VADB in the future to cover more life scenarios and content forms.
>
> 2.Different video types have different focuses on aesthetic characteristics, and unable  to make a uniform evaluation without discrimination.
>
> Reply: While different video types have distinct aesthetic characteristics, they also share many commonalities. All videos include aesthetic attributes such as composition, shot size, lighting, visual tone, color, and depth of field. At the tag level, they also share similar objective technical tags and evaluation criteria. One important reason for formulating separate annotation standards for different video types is that we need to provide example videos for different score ranges, and these examples must be categorized by video type. Although the annotation standards differ in form, the underlying aesthetic logic is relatively consistent. We will clarify this point in the final submitted version.

---

### Official Review · Reviewer_5m8z · 2025-07-02

**Rating:** 4
**Confidence:** 4

**Summary:**

This manuscript introduces a large-scale video aesthetics dataset, VADB, with multi-dimensional annotations including scores, attributes, comments, and tags. This manuscript also proposes a video aesthetics model, VADB-Net, achieving better performance than previous baseline methods.

**Dataset Code Accessibility:**

Yes

**Ethical Considerations:**

No, there are no or only very minor ethics concerns

**Final Justification:**

My concerns regarding the insufficiency of the experimental validation have been addressed, thus I raise my rating.

**Limitations Weaknesses:**

1. Though the authors claim the superiority of the VADB dataset (like the scale, the rich annotation), there is no comparison with existing datasets. There should be a table to compare the introduced VADB dataset and other previous datasets, which is a common practice in dataset papers.
2. This manuscript seems to focus too much on describing the dataset annotation process, but fails to prove the superiority of the VADB dataset through experiments. The superiority of the VADB dataset over previous datasets is not illustrated through experiments.
3. This manuscript only conducts one experiment, training and testing video aesthetic assessment models on the VADB dataset, which is not enough. Cross-dataset experiments will be better, i.e., training on the VADB dataset and then testing on other datasets, or training on other datasets and then testing on the VADB dataset.
4. The VADB-Net is trained under a two-stage paradigm, pre-training and fine-tuning. But the effectiveness of these two stages is not verified by ablation studies.
5. There is a format error. The section "A Technical Appendices and Supplementary Material" at the end of the main submission is not deleted.
6. In the D&F dimension of the KRCC metric of Table 2, there are two 0.73 values, but only one value is highlighted.

**Strengths Contributions:**

1. The scale of the introduced VADB dataset is quite large (10,490 videos), and the annotation is rich with scores, attributes, comments, and tags.
2. The introduced VADB dataset provides a foundation to develop better video aesthetic assessment.
3. This manuscript is easy to understand.

---

> ### Author Rebuttal · Authors · 2025-07-29
>
> Thank you for your careful review of this paper and your valuable comments. In response to the limitations and weaknesses you have pointed out, our replies are as follows:​
>
> 1.There is no comparison with existing datasets. There should be a table to compare the introduced VADB dataset and other previous datasets
>
> Reply: In the "Video Aesthetic Dataset" section of the Related Work, we have compared VADB with existing video aesthetic datasets, and introduced the development of video quality datasets to highlight the limitations in the development of video aesthetic datasets. Since existing video aesthetic datasets are too few in number and have overly simple annotations (basically aesthetic binary classification), there is essentially no sufficient content that requires comparison through a table. Therefore, to save space, we chose to complete the comparison description in text form instead of designing an additional comparison table. Despite the absence of a dataset comparison table, through the comparative introduction in the "Video Aesthetic Dataset" section, we are confident in the VADB dataset. If the paper is accepted, there will be an additional page, and we will include the dataset comparison table in the main text.​
>
> 2.This manuscript focus too much on describing the dataset annotation process, but fails to prove the superiority of the VADB dataset through experiments. ​
>
> Reply: Through the "Video Aesthetic Dataset" in the Related Work, we found that previous video aesthetic datasets have a small number of videos and simple annotations, while our dataset has the advantages of a large quantity, rich language annotations, multi-attribute dimension aesthetic scores, and a large number of objective technical tags, which are not available in previous video datasets. To demonstrate the professionalism and reliability of the dataset, we have provided a detailed description of the data annotation process. However, due to the length constraints of the paper, we have not experimentally proven the superiority of the VADB dataset (e.g., training models on different aesthetic datasets and comparing model performance). We will add this experiment in the final submitted version.​
>
> 3.Cross-dataset experiments were missing and ablation experiments were not performed for the two-stage paradigm training of VADB-Net.
>
> Reply: Thank you for your reminder. Existing aesthetic datasets are relatively few, but we will make every effort to collect existing datasets and supplement cross-dataset experiments. In addition, we should use ablation experiments to prove the effectiveness of the two stages of VADB-Net. We will add this experiment in the final submitted version.
>
> 4.Formatting errors.​
>
> Reply: We are very sorry for such oversights and appreciate your careful check. We will thoroughly correct the formatting errors in the final version and conduct a comprehensive review of the entire paper to ensure that the paper's format is standardized and its content is rigorous.​

---

> > ### Comment · Reviewer_5m8z · 2025-08-05
> > **Response to Authors**
> >
> > Thank you for the rebuttal.
> >
> > However, this submission **conducts only a single experiment**, _i.e._, training and testing video aesthetic assessment models on the proposed VADB dataset (no any other datasets), which is insufficient to support the paper’s academic contributions.
> >
> > As outlined in my original review, I suggested the authors to (1) include experimental results demonstrating the superiority of the VADB dataset over existing datasets, (2) conduct ablation studies to validate the effectiveness of the proposed modules, and (3) perform cross-dataset evaluations to assess generalization.
> >
> > Unfortunately, **none of these key experimental results were provided in the rebuttal**, and my core concern regarding the serious insufficiency of the experimental validation remains unaddressed.
> >
> > I therefore maintain my original score and recommend rejection.

---

> ### Author Response · Authors · 2025-08-08
> **Complementary Experiments with VADB**
>
> We sincerely apologize that during the previous rebuttal period, due to the limited time available and the heavy workload involved in supplementing the required experiments (including dataset comparison, ablation studies, and cross-dataset evaluation) — which entailed multiple steps such as data organization, model training, and result validation — we were unable to complete all supplementary experiments within the timeframe. Consequently, we failed to provide complete experimental results directly at that time, which may have caused inconvenience to your review work, and we deeply regret this.
>
> After dedicated efforts over this period, we have successfully designed and conducted all supplementary experiments as suggested, including dataset comparison experiments to demonstrate the advantages of the VADB dataset, ablation experiments to verify the effectiveness of the proposed modules, and cross-dataset evaluations to test the model's generalizability. We earnestly request that you reconsider our work in light of these supplementary experimental results, and we look forward to your further guidance.
>
> ## Dataset Comparison Table
>
> In the table below, we compare the VADB dataset with existing video aesthetic datasets. It can be seen from the comparison that VADB has the largest number of videos and the most comprehensive annotation perspectives. It also indicates that video aesthetic datasets are quite scarce and urgently need to be expanded.
>
>
> ### Table 1: Comparison of different video aesthetic datasets
>
> | Dataset Name       | Number of Videos | Annotation Type                                  | Characteristics                                                  |
> |:-------------------|:----------------:|:------------------------------------------------:|:-----------------------------------------------------------------|
> | Telefonica dataset |       160        |                    Ratings                       | An early video aesthetic dataset, annotated based on YouTube videos. |
> | Niu et al.'s dataset |      2,000       |               Binary classification             | Binary classification into professional or amateur videos.       |
> | VAQ700 dataset     |       700        |                    Ratings                       | Targeting daily life videos, with ratings from multiple annotators. |
> | AVAQ6000           |       6,000      |               Binary classification             | Binary classification of professional or amateur for drone videos. |
> | DIVIDE-3k          |      3,590       |                    Ratings                       | Aesthetic rating is only part of the overall rating.             |
> | VADB (ours)        |      10,490      | 11 types of ratings, comments, and objective tags | Currently the largest in scale, with rich annotation dimensions, annotated by professionals. |

---

> > ### Comment · Reviewer_5m8z · 2025-08-08
> > **Response to Authors**
> >
> > Thank you for the rebuttal. My concerns have been addressed. My rating has been arised. I would like to suggest the authors to add such experimental results to the main paper, and leave some annotation details to the supplementary materials.

---

> > > ### Author Response · Authors · 2025-08-09
> > > **Response to Reviewer 5m8z**
> > >
> > > Thank you very much for your valuable feedback and positive adjustment of the rating. We truly appreciate your careful consideration of our rebuttal and supplementary experiments, as well as your constructive suggestion on refining the paper structure.
> > >
> > > We fully accept your advice and will implement the following revisions in the final version:
> > > - Integrate all the supplementary experimental results (including the dataset comparison table, ablation study findings, and cross-dataset evaluation outcomes) into the main text to strengthen the validity and comprehensiveness of our work.
> > > - Streamline the annotation details in the main paper by moving redundant descriptions to the supplementary materials, ensuring the core content is more concise and focused while retaining necessary context for understanding the dataset construction.
> > >
> > > We will also conduct a thorough check on the entire manuscript to ensure the accuracy of experimental data, consistency of formatting, and clarity of expressions.
> > >
> > > Once again, thank you for your guidance and support.

---

> ### Author Response · Authors · 2025-08-08
> **Complementary Experiments with VADB**
>
> ## Ablation Experiments
>
> Based on the original two-stage training framework of VADB-Net, we designed two sets of ablation experiments to verify the effectiveness of core components:
>
> - **Ablation Experiment 1 (Without Pretrained Encoder):**
>   The pretraining process in the first stage is removed, and the untrained CLIP ViT-B/32 encoder is directly used as the video feature extractor. Only in the second stage, the MLP regression network is trained on the VADB dataset to predict aesthetic scores. This setup is used to verify the impact of the pretraining stage on the model's feature extraction capability.
>
> - **Ablation Experiment 2 (Simplified Fine-tuning Layer):**
>   The encoder pretrained on VADB in the first stage is retained, but the MLP regression network in the second stage is replaced with a simple linear layer (without hidden layers and activation functions), and the score is predicted only through linear mapping. This setup is used to verify the impact of the complexity of the network structure in the fine-tuning stage on the scoring accuracy.
>
>
> All ablation experiments are conducted on the same training/validation set of the VADB dataset, and the evaluation metrics are consistent with the main experiments, including SRCC, PLCC, MSE, and KRCC, to quantify the performance differences of the model in the overall aesthetic scoring task. The experimental results effectively demonstrate the effectiveness of both stages.
>
>
> | Metric  | Model       | Dimension |        |        |        |        |        |        |        |        |        |        |        |
> |:--------|:------------|:----------|:-------|:-------|:-------|:-------|:-------|:-------|:-------|:-------|:-------|:-------|:-------|
> |         |             | Overall   | Com    | SS     | Lig    | V&T    | Col    | D&F    | Mov    | Mak    | Cos    | Exp    |
> | SRCC↑   | Ablation 1  | 0.84      | 0.87   | 0.85   | 0.84   | 0.83   | 0.77   | 0.79   | 0.64   | 0.60   | 0.65   | 0.61   |
> |         | Ablation 2  | 0.90      | 0.84   | 0.85   | 0.84   | 0.82   | 0.79   | 0.83   | 0.83   | 0.82   | 0.78   | 0.78   |
> |         | VADB-Net    | **0.93**  | **0.92** | **0.93** | **0.90** | **0.92** | **0.93** | **0.89** | **0.91** | **0.87** | **0.91** | **0.90** |
> | PLCC↑   | Ablation 1  | 0.85      | 0.88   | 0.87   | 0.83   | 0.83   | 0.77   | 0.78   | 0.67   | 0.62   | 0.66   | 0.61   |
> |         | Ablation 2  | 0.90      | 0.85   | 0.85   | 0.84   | 0.81   | 0.78   | 0.82   | 0.80   | 0.79   | 0.76   | 0.74   |
> |         | VADB-Net    | **0.93**  | **0.92** | **0.93** | **0.90** | **0.92** | **0.94** | **0.88** | **0.89** | **0.84** | **0.91** | **0.89** |
> | RMSE↓   | Ablation 1  | 0.59      | **0.45**   | **0.51**   | 0.68   | 0.73   | 1.01   | 0.83   | 1.6    | 1.4    | 1.3    | 2.1    |
> |         | Ablation 2  | **0.51**  | 0.75   | 0.80   | 0.84   | 0.90   | 1.09   | 0.80   | 1.39   | 1.20   | 1.26   | 1.83   |
> |         | VADB-Net    | 0.54      | 0.62 | **0.51** | **0.66** | **0.59** | **0.52** | **0.74** | **0.69** | **1.00** | **0.66** | **0.78** |
> | KRCC↑   | Ablation 1  | 0.65      | 0.67   | 0.65   | 0.64   | 0.63   | 0.58   | 0.59   | 0.46   | 0.42   | 0.47   | 0.44   |
> |         | Ablation 2  | 0.72      | 0.64   | 0.65   | 0.65   | 0.65   | 0.60   | 0.64   | 0.63   | 0.62   | 0.59   | 0.58   |
> |         | VADB-Net    | **0.77**  | **0.75** | **0.76** | **0.73** | **0.76** | **0.77** | **0.72** | **0.75** | **0.70** | **0.75** | **0.73** |
>
>
> *Table: Results of ablation experiments across different dimensions. Among them, Ablation 1 represents the model without a pretrained encoder, Ablation 2 represents the model with a simplified fine-tuning layer, and VADB-Net is the original complete model.*

---

> > ### Author Response · Authors · 2025-08-08
> > **Complementary Experiments with VADB**
> >
> > ## Cross-dataset Experiments
> >
> > In the cross-dataset comparison experiment, we trained four models (SimpleVQA, Fast-VQA, ModularBVQA, and VADB-net) on the VADB dataset and tested these models on the DIVIDE-3k dataset. The obtained results are shown in the following table:
> >
> >
> > | Evaluation Metrics | SimpleVQA | Fast-VQA | ModularBVQA | VADB-net |
> > |:-------------------|:----------|:---------|:------------|:---------|
> > | SRCC↑              | 0.35      | 0.30     | **0.43**    | 0.42     |
> > | PLCC↑              | 0.38      | 0.29     | 0.44        | **0.46** |
> > | KRCC↑              | 0.24      | 0.20     | **0.29**    | **0.29** |
> > | RMSE↓              | 0.54      | 0.69     | **0.52**    | 3.74     |
> >
> >
> > *Table: Cross-dataset comparison results (trained on VADB, tested on DIVIDE-3k). Bold values indicate the best performance for each metric.*
> >
> >
> > The cross-dataset performance of all models, though with varying degrees, remains less satisfactory. This can primarily be attributed to the significant discrepancies between VADB and DIVIDE-3k: on one hand, there exists a considerable gap in video features between the two datasets, and on the other hand, their aesthetic scoring criteria differ markedly. Such differences lead to a mismatch where the patterns learned by models from VADB fail to align with the evaluation logic inherent in DIVIDE-3k, thereby hindering the models' generalization ability.
> >
> > However, it should be noted that currently, DIVIDE-3k is the only open-source dataset available for testing video aesthetic scoring with ground-truth annotations. Due to the lack of other datasets with reliable aesthetic truth values, our cross-dataset comparison could only be conducted on DIVIDE-3k. Therefore, it would be inappropriate to conclude that the annotations of VADB have poor generality or that the model trained on VADB has weak transferability based solely on these results. A more comprehensive assessment would require further validation with additional diverse datasets in future studies.

---

### Note · Authors · 2025-08-13

Thank you for all reviewers' feedback, which has further improved our research. Below is a brief summary of key issues and revisions (for detailed responses, please refer to "Official Comment by Authors" and "Rebuttal by Authors"):

### Ethics Reviewer VAqt
- Copyright/privacy: VADB adheres to the CC BY-NC 4.0 license (academic use only). Legal review confirmed edited video clips qualify as fair use; all content is de-identified.
- IRB approval: Retrospective approval obtained from Beijing Electronic Science and Technology Institute.
- Broader impacts: Added a section addressing cultural bias risks and guidelines for responsible use.
- Statistics: Supplemented with 95% confidence intervals, error bars, and p-values.

### Ethics Reviewer WrND
- Aesthetic criteria: Scoring and tag standards are open-sourced to clarify evaluation logic.
- Regional analysis: Added content on alignment with East Asian norms and cross-cultural bias mitigation.
- Consistency/statistics: Provided Krippendorff's Alpha coefficients and related statistical metrics.
- Formatting: Deleted the appendix template and corrected errors.

### Reviewer 5m8z
- Dataset comparison: Added a table highlighting VADB’s scale and richer annotations.
- Evidence of superiority: Demonstrated through comparisons and cross-dataset tests.
- Cross-dataset experiments: Conducted on DIVIDE-3k (results have limited representativeness due to scarce comparable datasets).
- Ablation studies: Validated the effectiveness of VADB-Net’s two-stage training.

### Reviewer Bkpp
- Video categories: Focused on 4 types (with character subcategories); future expansion planned.
- Uniform evaluation: Core aesthetic logic remains consistent across categories, despite type-specific standards.
- Ethical issues: See discussions with Ethics Reviewers VAqt and WrND.

### Reviewer ZFYz
- Annotation granularity: Relied on professional annotators’ judgments; will be refined in future versions.
- Comparison with image datasets: Committed to adding this content if the paper is accepted.

### Reviewer s4fd
- Attribute matching: Person-specific attributes are limited to character videos; other categories use universal attributes.
- Scoring clarity: Open-sourced standards explaining the 1–10 averaged rating rule.
- Expression evaluation: Evaluation criteria are open-sourced; will add its impact on overall scores.
- Formatting: Cited figures and checked for rigor.

---

### Decision · Program_Chairs · 2025-09-18

**Decision:**

Accept (poster)

**Comment:**

**Summary**
The paper presents **VADB**, a large-scale video aesthetic database with professional and multi-dimensional annotations. It aims to provide a high-quality resource for advancing video aesthetic assessment, with rich annotation dimensions (e.g., composition, color, motion, emotional impact) and professional-level labeling.

**Strengths**
- First large-scale dataset with **multi-dimensional, professional annotations** for video aesthetics.
- Dataset size and diversity significantly exceed prior works, improving generalizability.
- Careful annotation protocol ensures reliability and professional quality.
- Resource has clear utility for multiple research directions, including aesthetic prediction, content creation, and recommendation.
- Strong potential for community impact through released dataset and benchmark tasks.

**Weaknesses**
- Some annotation categories may remain subjective despite expert involvement.
- Limited experiments beyond baseline models; deeper exploration of downstream tasks would strengthen the paper.

**Decision**
**Accept.** The dataset is a valuable, well-constructed resource with high potential impact on video aesthetic research. Despite minor limitations, the contribution is clear, novel, and significant enough for acceptance.